# Phosphatase activity tunes two-component system sensor detection threshold

Brian P. Landry [1], Rohan Palanki[1], Nikola Dyulgyarov[1], Lucas A. Hartsough[1] & Jeffrey J. Tabor [1,2]

Two-component systems (TCSs) are the largest family of multi-step signal transduction pathways in biology, and a major source of sensors for biotechnology. However, the input concentrations to which biosensors respond are often mismatched with application requirements. Here, we utilize a mathematical model to show that TCS detection thresholds increase with the phosphatase activity of the sensor histidine kinase. We experimentally validate this result in engineered *Bacillus subtilis* nitrate and *E. coli* aspartate TCS sensors by tuning their detection threshold up to two orders of magnitude. We go on to apply our TCS tuning method to recently described tetrathionate and thiosulfate sensors by mutating a widely conserved residue previously shown to impact phosphatase activity. Finally, we apply TCS tuning to engineer *B. subtilis* to sense and report a wide range of fertilizer concentrations in soil. This work will enable the engineering of tailor-made biosensors for diverse synthetic biology applications.

[1] Department of Bioengineering, Rice University, 6100 Main St., Houston 77005 TX, USA. [2] Department of Biosciences, Rice University, 6100 Main St., Houston 77005 TX, USA. Correspondence and requests for materials should be addressed to J.J.T. (email: jeff.tabor@rice.edu)

central goal of synthetic biology is to program cells to sense and respond to chemical or physical inputs in desired ways[1]. To this end, researchers develop genetically encoded sensors, often based upon multi-step signal transduction pathways or one-component transcription factors[2] that convert inputs of interest into biological signals such as gene expression. However, all biosensors respond to their cognate inputs over finite concentration ranges that are often mismatched with application demands[3].

Despite this challenge, there has been little focus on developing technologies for tuning biosensor detection windows. In two recent studies, the input concentrations required to activate *Escherichia coli* nitrate and hydrogen peroxide sensors by 50% (i.e., the detection thresholds, quantified by the parameter $K_{1/2}$) were decreased 412- and 15-fold by linking the respective sensors to the expression of a phage recombinase that inverts a segment of DNA into an orientation appropriate for transcription of an output gene[4,5]. Though this approach is simple and modular, the recombination step is irreversible and delays sensor response by up to 15 h, making it incompatible with applications requiring dynamic or rapid responses. In a separate pair of yeast studies, RNA secondary structure design was used to lower the detection threshold of an engineered theophylline-responsive antiswitch from 10 mM to 1 mM[6], and protein expression level optimization was used to reduce the estradiol detection threshold of the mitogen-activated protein kinase (MAPK)/extracellular signal-regulated kinase pathway from 32 μM to 6.6 μM[7]. However, antiswitches currently sense a limited number of inputs, and both of these approaches yield modest changes in detection threshold, limiting the utility of these strategies. Finally, computational design[8] and directed evolution[9] of ligand-binding transcription factors show promise for tuning sensor detection thresholds. However, these methods are time and labor intensive and require extensive domain-specific expertise, limiting their widespread use.

Two-component systems (TCSs) are an important source of sensors for synthetic biology. Tens of thousands of TCSs have been identified in bacterial genome sequences. Individual members of this family sense inputs as diverse as metal ions of particular oxidation states[10], respiratory electron acceptors[11], gases[12], inorganic phosphate[13], heme[14], quorum sensing autoinducers[15], antimicrobial peptides[16], simple sugars[17], gut polysaccharides derived from the diet[18] or host[19], human[20] and plant[21] hormones, oxidative stress[22], physical contact[23], and

specific wavelengths of light[24]. Synthetic biologists have begun to repurpose light-sensing TCSs to function as sensors for optogenetics[25–28] and chemical-sensing TCSs to engineer diagnostic gut bacteria[29–31], among other applications.

The prototypical TCS comprises two proteins: a sensor histidine kinase (SK) and a response regulator (RR) (Fig. 1a). The SK contains a (typically extracellular) N-terminal sensor domain that switches from an inactive to active conformation in the presence of the input[32]. This conformational change is transmitted to a C-terminal cytoplasmic signaling region comprised of catalytic and adenosine triphosphate (ATP) binding (CA) and dimerization and histidine phosphotransfer (DHp) domains. The CA domain catalyzes the transfer of the gamma phosphoryl group from ATP to a conserved histidine residue within the DHp domain. The phosphorylated SK (SK~P) binds the RR via a DHp interaction interface, and transfers the phosphoryl group to a conserved RR aspartate. Phosphorylation activates the RR, driving it to modulate transcription from one or more output promoters. Many SKs are also bi-functional and dephosphorylate the phosphorylated RR (RR~P) (Fig. 1a)[33]. The presence of input increases the rate at which the RR is phosphorylated, decreases the rate at which the RR~P is dephosphorylated, or both[33]. Many SK mutations, in both the DHp and CA domains, have been identified that decrease this phosphatase activity, resulting in increased RR~P levels[34–39]. When this increase is substantial, it results in leaky transcriptional output, i.e., output in the absence of input[35,36,39]. However, the impact of these phosphatase-altering mutations on TCS detection thresholds has not been considered.

Here, we combine mathematical modeling with an experimental synthetic biology approach to show that mutations that alter SK phosphatase or kinase activity can be used to rationally tune TCS detection thresholds. We demonstrate that our method functions in Gram-negative and Gram-positive bacteria and in diverse chemical-sensing TCSs. We go on to demonstrate that a widely conserved residue can be mutated to tune the detection thresholds of two recently described TCSs for which signaling mutations have not yet been identified. Finally, we utilize *Bacillus subtilis* expressing wild-type and sensitivity-enhanced nitrate sensors to quantify a wide range of fertilizer levels in soil. These sensors could be used to control the expression of engineered nitrogen fixation pathways in order to achieve synthetic nitrate homeostasis in soil.

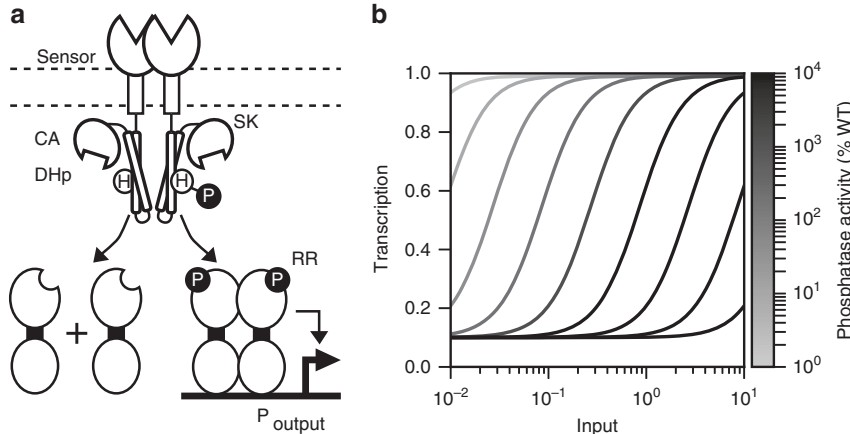

**Fig. 1** Mathematical model reveals that SK phosphatase activity tunes TCS detection threshold. **a** Diagram of a canonical TCS. **b** Model simulations of the relationship between TCS input concentration and transcriptional output rate (i.e., transfer function) wherein SK phosphatase activity is varied between 1% and 10,000% of wild-type (Supplementary Note 1). Detection threshold ($K_{1/2}$), or the input concentration where transcriptional output is half-maximal, increases with phosphatase activity. There is no trade-off between detection threshold and dynamic range for intermediate changes in phosphatase activity; however, a trade-off emerges for strong changes in phosphatase activity (Supplementary Fig. 1)

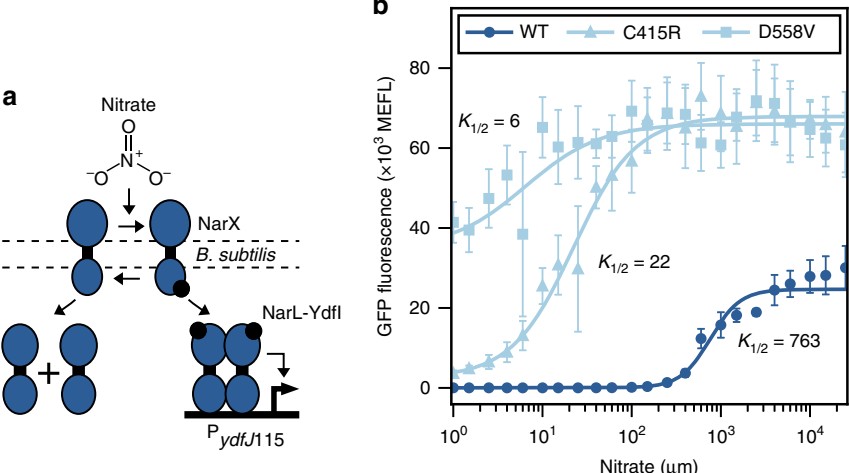

**Fig. 2** Phosphatase-reducing mutations decrease detection threshold in a *B. subtilis* nitrate sensor. **a** An engineered *B. subtilis* nitrate-sensing TCS. The *E. coli* SK NarX signals to a chimeric RR composed of the N-terminal Receiver (REC) domain of *E. coli* NarL fused to the C-terminal DNA binding domain (DBD) of *B. subtilis* YdfI (Supplementary Fig. 2). Phosphorylated NarL-YdfI activates transcription from the *B. subtilis* $P_{ydfJ}115$ output promoter. **b** Transfer functions of variants from (**a**) containing wild-type NarX, NarX(C415R), and NarX(D558V). Points represent the mean and error bars the standard error of the mean (SEM) of experiments on three separate days

## Results

**Mathematical model of TCS detection threshold**. We hypothesized that TCS detection thresholds could be tuned by introducing mutations that alter SK kinase or phosphatase activity without compromising the overall response (i.e., dynamic range, or ratio of output in saturating versus zero input) of the system. Specifically, we considered that the detection threshold of a TCS occurs at the particular RR~P concentration that elicits a half-maximal output promoter response (i.e., RR~$P_{1/2}$). For any input concentration, the corresponding RR~P concentration is set by the ratio of SK kinase to phosphatase activity[40]. Thus, we reasoned that mutations that enhance kinase or reduce phosphatase activity should result in RR~$P_{1/2}$ being reached at a lower input concentration, thereby reducing TCS detection threshold. The opposite should also be true: TCS detection thresholds should increase with kinase-reducing or phosphatase-enhancing mutations. Furthermore, if a mutation is sufficiently weak that the window of altered RR~P concentrations still traverses the range to which the output promoter is sensitive, there should be little effect on TCS dynamic range.

To examine this hypothesis, we first utilized a previous mathematical model of TCS signaling[41]. We parameterized the model with the best available in vivo experimental values of TCS reaction rates as determined for the well-studied inorganic phosphate-sensing TCS PhoRB[42]. Then, we set the phosphatase activity parameter to different values between 1% and 10,000% that of wild type. Finally, we evaluated the resulting detection thresholds by simulating the relationship between input concentration and gene expression output (i.e., the transfer function) in each case (Supplementary Note 1)[43]. In agreement with our hypothesis, the model predicts that TCS detection threshold can be tuned by altering SK phosphatase activity (Fig. 1b). Moreover, intermediate changes in phosphatase activity alter detection threshold without impacting dynamic range (Supplementary Fig. 1). As expected, large decreases or increases in phosphatase activity result in high basal or low maximal expression, respectively, and thereby reduce dynamic range.

We also found that modulating kinase activity had the reciprocal effect to that of modulating phosphatase activity, with increasing kinase activity decreasing the detection threshold and decreasing kinase activity increasing the detection threshold

(Supplementary Fig. 1). However, our primary goal is to decrease TCS detection thresholds, and it is easier to identify mutations that decrease rather than increase enzymatic activity. Thus, we chose to focus on decreasing phosphatase activity as opposed to increasing kinase activity. This decision is supported by mutational screens of SK activity that have found that decreases in phosphatase activity are much more common than increases in kinase activity[38].

**Tuning the detection threshold of a nitrate sensor**. To examine our modeling results experimentally, we selected two point mutations, C415R and D558V, that decrease the phosphatase activity of the *E. coli* nitrate-activated SK NarX via different mechanisms and to different extents. C415R targets the DHp interaction interface, weakens the interaction between NarX and its cognate RR NarL, and causes a moderate reduction in phosphatase activity[39]. On the other hand, D558V targets the CA domain and is thought to decrease phosphatase activity more strongly than C415R. However, because its impact has been measured only with gene expression assays, it is also possible that D558V may increase kinase activity[39]. We measured the nitrate detection thresholds of a wild-type NarXL that we engineered to function in *Bacillus subtilis*, and its corresponding C415R and D558V variants (Fig. 2a; Supplementary Fig. 2). The wild-type system exhibits a relatively high $K_{1/2}$ of 762 μM (95% confidence interval (CI) 629–963 μM) (Fig. 2b). On the other hand, the medium strength C415R mutation decreases the value substantially ($K_{1/2} = 22$ μM, 95% CI 16–33 μM), and the strong D558V mutation reduces it even further ($K_{1/2} = 6$ μM, 95% CI 0–23 μM) (Fig. 2b).

Dynamic range is commonly reported as the primary performance metric for biosensors. The C415R and D558V versions of our nitrate sensor exhibit decreased dynamic range due to increased minimum output levels (Fig. 2b). Thus, we individually optimized SK and RR expression levels in these mutated sensors in an effort to maximize the dynamic range for each (Supplementary Fig. 3). Consistent with our modeling results, maximal dynamic range decreases from 1909-fold (wild type), to 78-fold (C415R) and 2-fold (D558V) (Supplementary Figs. 1, 3). On the other hand, the amplitude range, or difference between maximum and minimum output, may be a more useful

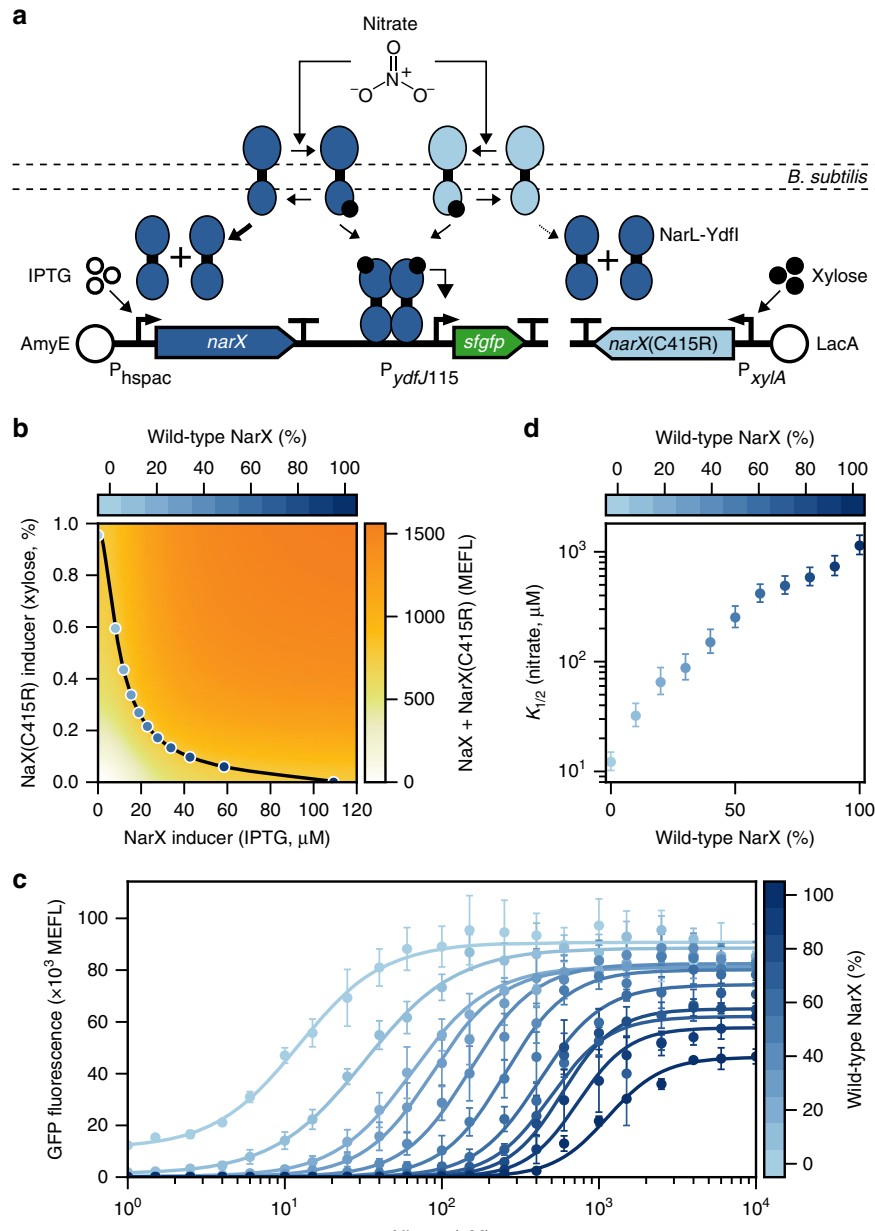

**Fig. 3** Detection threshold decreases continuously with phosphatase activity. **a** Cartoon of the genetic system engineered for the iso-SK experiment. Both NarX and NarX(C415R) are expressed under inducible promoters in the same cell. Both wild-type and mutant SKs have kinase and phosphatase activity on a constitutively expressed NarL-YdfI. **b** IPTG and xylose are used to express different levels of NarX and NarX(C415R) resulting in different total SK concentrations (Supplementary Figure 6). We selected induction levels that result in 775 MEFL (Methods) of total SK expression (black line), while evenly varying the percentage of NarX compared to total SK levels between 0 and 100% (blue points). Requisite inducer concentrations were calculated by inverting the Hill fits $\left([\text{inducer}] = \sqrt[n]{\frac{r}{1-r}}, \text{where } r = \frac{[\text{SK}]-\text{low}}{\text{high}-\text{low}}\right)$ in (Supplementary Fig. 6). **c** Nitrate transfer functions for the iso-SK strain when induced with IPTG and xylose to the 11 different percentages of wild-type NarX from (**b**). Points represent the mean and error bars the standard error of the mean (SEM) of experiments on three separate days. **d** The relationship between the percent NarX and the $K_{1/2}$ of the Hill function fits in (**c**). Points represent the $K_{1/2}$ values and error bars the 95% confidence intervals of the best fits

performance metric for many applications. While the amplitude range of our wild-type nitrate sensor is 24,652 molecules of equivalent fluorescein (MEFL) (21 MEFL to 24,664 MEFL), it increases to 65,402 MEFL (2,508 MEFL to 67,910 MEFL) for C415R and 31,294 MEFL (34,758 MEFL to 66,052 MEFL) for D558V (Fig. 2b). These results provide compelling initial support for our approach.

To more rigorously validate TCS tuning, we next developed a strategy to continuously vary phosphatase activity in live cells (Fig. 3a). Specifically, we expressed wild-type NarX and NarX

(C415R) under two different chemically inducible promoters and utilized green fluorescent protein (GFP) fusions and quantitative flow cytometry to map the relationship between inducer and SK levels (Fig. 3b; Supplementary Figs. 4–6). Then, we used different inducer combinations to achieve NarX/NarX (C415R) expression ratios between 0% and 100% at a constant total SK expression level (NarX+NarX (C415R)) (Fig. 3b). Assuming NarX and NarX (C415R) function identically outside of their different phosphatase activities, tuning their expression ratio in this way enables us to continuously vary phosphatase activity between mutant and

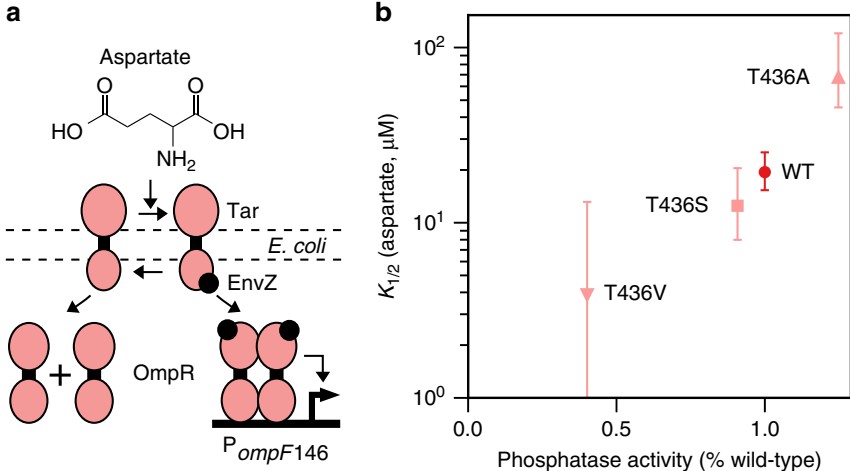

**Fig. 4** TCS tuning an *E. coli* aspartate sensor. **a** The Taz-OmpR aspartate sensing TCS in *E. coli*. The Taz SK is composed of the Tar sensing domain fused to the EnvZ DhP and CA domains. Taz signals to OmpR which controls transcription from the P*ompF* output promoter. **b** The relationship between the phosphatase activity of the mutant as measured by Inouye and colleagues[36] and the $K_{1/2}$ fit parameter from transfer function fits in Supplementary Fig. 5. Error bars are the 95% confidence interval of the $K_{1/2}$ fit parameter

wild-type levels. In strong agreement with our modeling results (Fig. 1b), the nitrate detection threshold decreases continuously from $K_{1/2} = 1138\ \mu M$ to $K_{1/2} = 12\ \mu M$ as the percentage of wild-type NarX decreases from 100 to 0% (Fig. 3c, d). The amplitude range increases from 46,372 MEFL (20 MEFL to 46392) to 87,324 (1,205 MEFL to 88,529 MEFL) as the percentage of mutant SK increases from 0 to 90%. Upon continued increase to 100% NarX (C415R), the amplitude range decreases slightly to 79,628 MEFL (11,118 MEFL to 90,746 MEFL) (Fig. 3c). We also observe that an eightfold decrease in detection threshold can be achieved with only a twofold decrease in the dynamic range, and this follows model predictions that moderate changes to the detection threshold have minor effects on dynamic range (Supplementary Note 1; Supplementary Figs. 1, 5). However, the large 100-fold decrease in detection threshold between 100% and 0% wild-type expression also decreases the dynamic range from 2,334-fold to 8-fold. This experiment clearly shows that TCS detection threshold can be tuned by tuning SK phosphatase activity. Furthermore, this iso-SK technique provides a synthetic biology method for tuning the detection threshold of a TCS to intermediate values not achievable using a mutation alone.

**Detection threshold tuning of an *E. coli* aspartate sensor.** To evaluate the extensibility of our technology to other sensors and organisms, we next examined the engineered *E. coli* aspartate-activated TCS Taz-OmpR (Fig. 4a). Here, the SK Taz phosphorylates and dephosphorylates the transcription-regulating RR OmpR. Taz phosphatase activity is high in the absence of aspartate, and low in its presence[44]. A previous study identified numerous phosphatase-altering mutations of different strengths at Taz T436. In particular, substituting S, V, E, D, and K at this site decreases phosphatase activity by 10, 60, 91, 91, and 98%, and introducing A increases phosphatase activity by 25%[36]. Consistent with our NarX results (Fig. 2), T436S and V reduce the Taz-OmpR aspartate detection threshold in proportion to their strength (wild type: 19 μM (95% CI 15–25 μM); T436S: 12 μM (95% CI 8–20 μM); T436V: 4 μM (95% CI 0–13 μM)) (Fig. 4b; Supplementary Fig. 7). Furthermore, T436A increases the detection threshold to 67 μM (95% CI 45–120 μM) (Fig. 4b; Supplementary Fig. 7). This T436A result indicates that the SK phosphatase activity alone, as opposed to an alternate effect of phosphatase-reducing mutations, is responsible for tuning TCS detection threshold. Furthermore, these data agree with previous

results that show drastic decreases in phosphatase activity result in lowered dynamic ranges, while smaller changes, such as with T436A, have little effect on dynamic range (Supplementary Note 1; Supplementary Fig. 7). We conclude that TCS tuning can be used to both reduce and increase detection threshold, and can be applied to diverse TCS sensors and host bacteria.

Interestingly, the strong T436E, D, and K mutations abolish the Taz-OmpR aspartate response altogether (Supplementary Fig. 7). Simultaneous introduction of C415R and D558V into NarX destroys signaling to NarL as well (Supplementary Fig. 8). These results demonstrate that if phosphatase mutations are too strong, the SK will fail to signal to the RR, thereby imposing limits on the magnitude of sensitivity enhancement.

**Bioinformatic identification of a TCS hot spot residue.** Unlike the initial model systems that we examined, most TCSs lack known phosphatase mutations. Therefore, we next aimed to develop a general method to apply TCS tuning to a wide range of systems. Taz T436 resides in the second (variable) position of the well-studied CA domain GXGXG motif, which is involved in binding an adenosine diphosphate co-factor that regulates SK phosphatase activity, as well as binding the phosphodonor ATP[32]. We performed a bioinformatic analysis that revealed that GXGXG is present in 64% of all bacterial SKs (Fig. 5a; Supplementary Fig. 9). Therefore, we hypothesized that the second GXGXG position might serve as a general hot spot residue that can be mutated to alter the detection thresholds of many TCSs.

To validate this strategy, we examined the tetrathionate sensor TtrSR and the thiosulfate sensor ThsSR (Fig. 5b, c; Supplementary Figs. 10, 11), two TCSs that we recently discovered in the genomes of marine *Shewanella* and ported into *E. coli*[30]. Like most SKs, the corresponding SKs TtrS and ThsS both contain the GXGXG motif and lack known phosphatase mutations. Therefore, we performed saturation mutagenesis on the second GXGXG residue in each (TtrS L627, ThsS L547) (Fig. 5a), and measured the response of both the wild-type and all 38 mutant TCSs to their cognate ligands (Supplementary Figs. 10, 11). Remarkably, we observed that 14 and 9 amino acids result in functional TtrSR and ThsSR sensors, respectively (Supplementary Figs. 10, 11). Most of the functional residues have high hydropathy scores, suggesting this site best tolerates hydrophobic amino acids (Supplementary Figs. 10, 11). Then, we characterized the transfer functions of the ten TtrSR and ThsSR variants

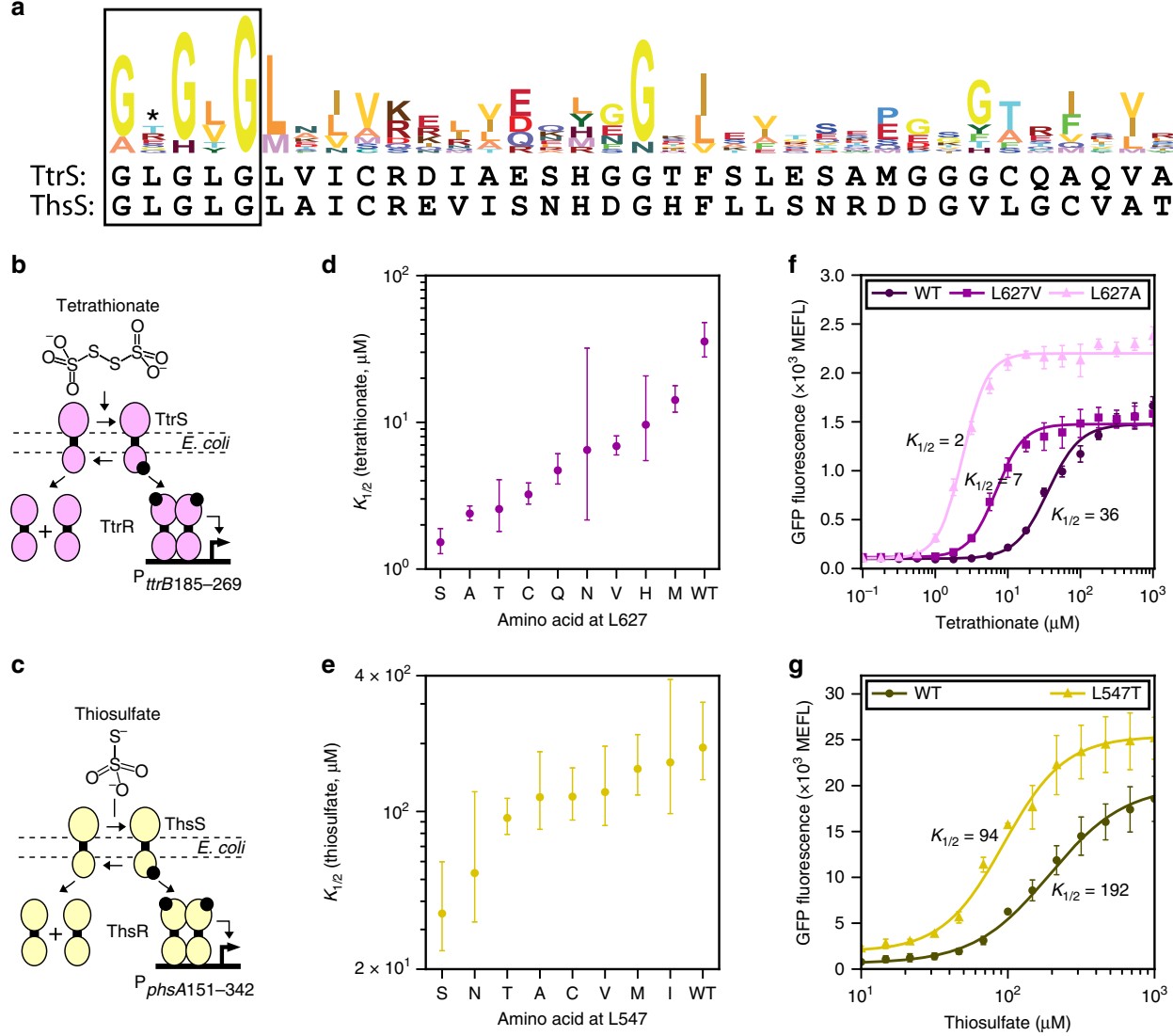

**Fig. 5** Identification of a phosphatase hot spot residue for TCS tuning. **a** A hidden Markov model (HMM) logo of the SK G2 box (Supplementary Fig. 9), which contains the GXGXG motif (black box). The phosphatase hot spot residue is indicated with an asterisk. The G2 boxes of TtrS and ThsS are shown below the HMM logo. Cartoons of *E. coli* TtrSR (**b**) and ThsSR (**c**). $K_{1/2}$ values of TtrSR (**d**) and ThsSR (**e**) wherein each SK contains the indicated amino acid at the phosphatase hot spot. Points represent the $K_{1/2}$ values and error bars the 95% confidence intervals of best fits to replicates on three different days (Supplementary Figs. 10, 11). **f**, **g** Transfer functions of selected TtrSR and ThsSR systems from (**d**, **e**). Points represent the mean and error bars the SEM of replicates on three separate days

exhibiting the largest fold activation (Supplementary Figs. 10, 11). All of the mutations that we tested lower the detection threshold (Fig. 5d–g). In the case of TtrSR, $K_{1/2}$ varies between 35.6 μM (95% CI 27–48 μM) for wild-type and 1.5 μM (95% CI 1.3–1.9 μM) for the strongest mutant (Fig. 5d). For ThsSR, $K_{1/2}$ varies between 192 μM (95% CI 138–305 μM) and 35 μM (95% CI 24–60 μM) (Fig. 5e). Because the CA domain is involved in the kinase and phosphatase reactions, further characterization is needed to determine which enzymatic activity, or activities, have been changed by these GXGXG mutations. Interestingly, we found that the TtrS(L627A) mutant not only decreased the detection threshold from 35.6 μM to 2.4 μM, but it also increased the dynamic range from 15- to 21-fold and the amplitude range from 1377 MEFL (100 MEFL to 1477 MEFL) to 2095 MEFL (105 MEFL to 2200 MEFL) (Fig. 5f). Conversely, decreasing the detection threshold of the thiosulfate sensor twofold with L547T resulted in a decrease in dynamic range from 34- to 13-fold and an increase in amplitude range from 19,390 MEFL (596 MEFL to

19,986 MEFL) to 23,482 MEFL (1905 MEFL to 25,387 MEFL) (Fig. 5g). We conclude that mutating the second GXGXG residue is a simple strategy for tuning the detection thresholds of diverse TCSs.

**Application of TCS tuning to fertilizer biosensing.** Finally, we set out to demonstrate a proof-of-principle application for TCS tuning. Nitrate is the primary source of nitrogen used by crops, and a major component of fertilizer. However, over-application of fertilizer causes billions of dollars in damage per year to human health and the environment[45]. Recently, synthetic biologists have expressed bacterial nitrogen fixation pathways, which ultimately convert atmospheric $N_2$ into nitrate, in non-native host bacteria[46]. However, heterologous production of nitrogen fixation pathways in soil bacteria could also lead to nitrate over-production. To prevent this outcome, genetic feedback control systems wherein bacteria sense a wide range of soil nitrate levels

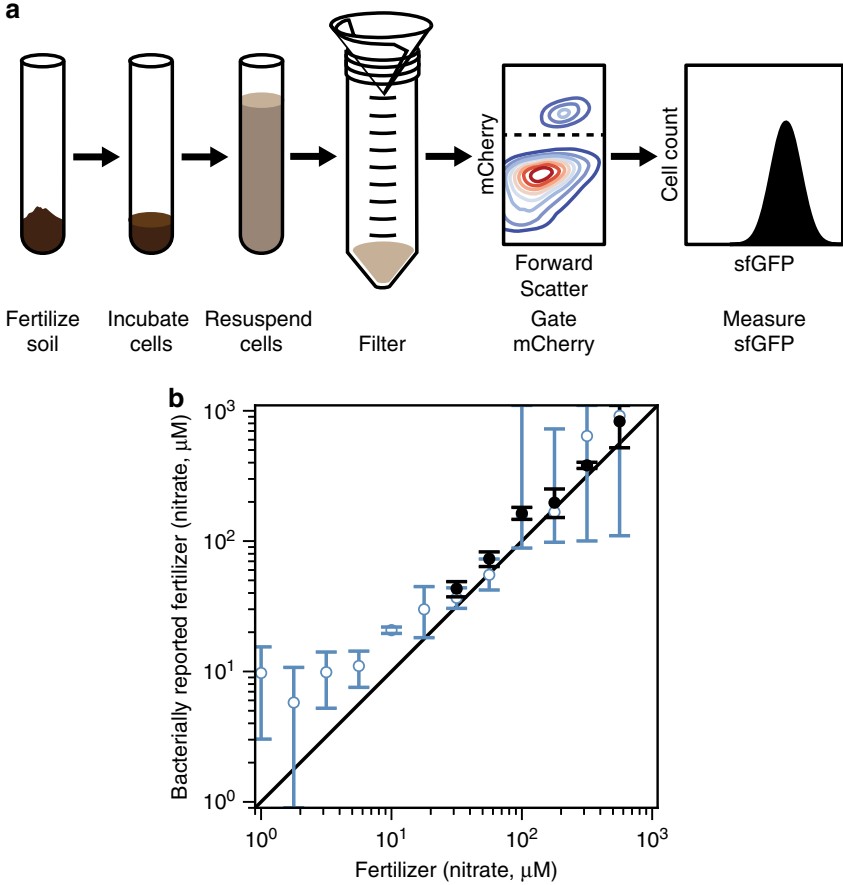

**Fig. 6** TCS tuning expands the detection range of an engineered soil fertilizer biosensor. **a** Experimental protocol for the incubation of bacteria with fertilized soil, subsequent purification via filtration, selection by mCherry fluorescence, and measurement of sfGFP levels (Methods). **b** Soil was fertilized with different amounts of nitrate as specified by the manufacturer. The level of fertilization was then measured with a *B. subtilis* strain expressing the NarXL-YdfI TCS with wild-type (black) and C415R (blue) NarX variants (Fig. 2a; Supplementary Fig. 12). This was compared to a previously measured standard curve of nitrate response to calculate a bacterially reported nitrate concentration from the fertilizer (Supplementary Note 2; Supplementary Fig. 12). The relationship between the actual and bacterially reported values is shown. Points represent the mean and error bars the SEM of experiments on three separate days

and induce nitrogen fixation pathways only to the extent that they are needed are highly desirable.

To demonstrate such a sensing capability, we incubated *B. subtilis* engineered to express our wild-type and C415R NarXL systems in soil spiked with various amounts of a nitrate standard, and measured the resulting superfolder GFP (sfGFP) fluorescence values via flow cytometry (Methods; Fig. 6a; Supplementary Fig. 12). Then, we used the resulting data to generate a standard curve relating sfGFP fluorescence to soil nitrate concentration (Supplementary Fig. 12; Supplementary Note 2). Then, we added different amounts of commercial fertilizer, rather than nitrate, to the soil (Methods; Supplementary Fig. 12). Using the standard curves, we compared the amount of nitrate reported by each of our sensor systems to the amount specified by the manufacturer (Supplementary Note 2). Indeed, the wild-type NarXL system enables estimation of fertilizer levels within twofold of the manufacturer value between the tested values of 31.6 μM and 562 μM nitrate, while the C415R system allows accurate detection between 5.62 μM and 562 μM (Fig. 6b). This experiment demonstrates that we can use TCS tuning to engineer bacteria to sense a large range of nitrate concentrations in a complex soil environment. Such broad-range sensing could be coupled with nitrogen fixation pathways to maintain soil nitrate at ideal levels in different agricultural contexts.

## Discussion

This work extends a growing suite of techniques for engineering TCSs to function as sensors for synthetic biology. First, literature searches[27,47,48] or bioinformatics[30] can be used to identify TCSs that sense inputs of interest. If a candidate TCS has a known output promoter, and functions in the desired host and environmental conditions, it can be used as an off-the-shelf sensor without further modifications[47,49]. Otherwise, the sensor domain can potentially be swapped onto the SK of a second TCS that contains a reliable output promoter, resulting in the design of a chimeric sensor[25,50,51]. Like all gene regulatory systems, TCSs can exhibit substantial 'leakiness' in the off state, or modest dynamic range. These performance features can be improved by redesigning the sequence of the output promoter and optimizing the expression levels of the SK and RR[27,30,48,52].

However, this workflow may produce sensors that do not respond appropriately to application-relevant input concentrations. For example, tetrathionate was previously shown to be elevated in the mouse colon during *Salmonella typhimurium*-induced inflammation[53]. Following this report, Silver and colleagues[31] used *S. typhimurium* TtrSR to activate a transcriptional memory circuit in order to engineer a gut bacterium that senses and remembers tetrathionate exposure in order to diagnose colon inflammation. However, despite 100% tetrathionate activation in vitro, most bacteria expressing this sensor device are not

activated by inflammatory conditions in vivo[31]. One possible reason for this discrepancy is that in vivo tetrathionate concentrations do not reach the *S. typhimurium* TtrSR detection threshold. Thus, by using TCS tuning to lower the detection threshold of TtrSR (Fig. 5f), it is possible that the performance of this diagnostic gut bacterium could be improved.

It is possible that nature uses phosphatase activity as a knob to tune TCS detection threshold as well. First, there are a wide range of SK residues that can be mutated to specifically alter phosphatase activity[39]. This fact suggests that evolution can tune TCS detection thresholds, which could enable organisms to adapt to new niches with different input concentrations. Interestingly, few mutations have been discovered that increase phosphatase activity, or decrease kinase activity. As of currently, this fact restricts our TCS tuning method to applications where lower detection thresholds (i.e., increases in sensitivity) are needed. However, sensitivity decreases are also desirable in many synthetic biology applications, which motivates future work to identify appropriate mutations.

Additionally, some SKs interact with phosphatase-modulating auxiliary proteins[54]. It is possible that these auxiliary proteins can tune the detection thresholds of the corresponding TCSs. Unlike SK mutations, they could also be dynamically induced or repressed in response to changing environmental or physiological conditions to temporarily adjust detection thresholds. This phenomenon is analogous to our use of chemically inducible promoters to adjust the NarXL nitrate detection threshold in our iso-SK experiment (Fig. 3). These intriguing possibilities remain to be explored.

Finally, our approach may be extensible to other kinase pathways. For example, eukaryotes use MAPK cascades to sense and respond to important extracellular signals such as growth factors and immunomodulators[55]. Threonine and tyrosine phosphatases modulate signaling through these pathways by dephosphorylating MAP kinases[56]. Researchers have expressed variants of these phosphatases under synthetic feedback control to re-program pathway response dynamics[57,58]. Alternatively, by constitutively expressing such phosphatases to different extents, or expressing phosphatases of different strengths, the detection thresholds of MAPK cascades could potentially be tuned.

In conclusion, we have demonstrated a simple, general strategy for tuning the detection threshold of TCSs—one of the largest and most diverse families of sensors in biology. Due to its effectiveness and ease of use, our method should have widespread applications in synthetic biology.

## Methods

**DNA and bacterial strain construction.** Details of synthetic DNAs used in this work are given in Supplementary Data 1-4. All *E. coli* systems are expressed on extrachromosomal plasmids. All plasmids were assembled via Golden Gate cloning[59]. Assembled plasmids were transformed into *E. coli* NEB 10-β (New England Biolabs, cat no. C3019H). Ribosome binding site (RBS) strengths were calculated using the RBS calculator[60].

All *B. subtilis* systems are constructed as linear double-stranded DNA Integration Modules (IMs) and integrated into the chromosome. All IMs were assembled with Golden Gate cloning[59]. Assembled DNA was amplified with PCR, transformed into *B. subtilis* 168 (BGSCID 1A1) and recombined into the chromosome using the two-step transformation protocol[61]. *B. subtilis* genomic DNA was then purified (Promega, A1120) and used for subsequent transformations.

*E. coli* NEB 10-β and *B. subtilis* 168 were grown in LB Miller broth shaking at 250 rpm at 37 °C. Then, 50 µg mL⁻¹ ampicillin, 35 µg mL⁻¹ chloramphenicol, and 100 µg mL⁻¹ spectinomycin for *E. coli* and 100 µg mL⁻¹ spectinomycin, 0.5 µg mL⁻¹ erythromycin, 5 µg mL⁻¹ chloramphenicol, and 5 µg mL⁻¹ kanamycin for *B. subtilis* were added where appropriate. Transformed strains were stored in 15% glycerol stocks at −80 °C.

*E. coli* plasmids are available from Addgene using accession numbers listed in Supplementary Data 3. *B. subtilis* constructs are available from the Bacillus Genetic Stock Center using BGSC numbers listed in Supplementary Data 4.

**In vitro nitrate experiments.** In vitro nitrate induction experiments were conducted with *B. subtilis* 168 Δ*ydfHI::camR* (iND46; Supplementary Fig. 2). C minimal media with sodium succinate and potassium glutamate (CSE media) containing 30 mM KH₂PO₄ (Fisher BioReagents, BP362-1), 70 mM K₂HPO₄ (Fisher BioReagents, BP363-1), 25 mM (NH₄)₂SO₄ (Sigma, A4418-100G), 10 mM MnSO₄ (Sigma-Aldrich, M7634-100G), 500 µM MgSO₄ (VWR, BDH9246-500G), 12.5 µM ZnCl₂ (Sigma, Z0152-50G), 245 µM L-tryptophan (Sigma-Aldrich, T0254-25G), 22 mg L⁻¹ ammonium iron(III) citrate (Sigma-Aldrich, F5879-100G), 43.2 mM Potassium Glutamate (Alfa Aesar, A17232), 22.2 mM Sodium Succinate (Alfa Aesar, 33386), and 43.4 mM Glycerol (Fisher BioReagents, BP229-1) were used without antibiotics. Induction conditions were 25 mM NaNO₃ (Sigma-Aldrich, S5506), 10 µM isopropyl β-D-1-thiogalactopyranoside (IPTG) (IBI Scientific, IB02125), and 1% xylose (Alfa Aesar, A10643) unless otherwise noted. IPTG and xylose levels were chosen for optimal fold change of the NarX(D558V) TCS (Supplementary Fig. 3). An overnight culture was inoculated from a 15% glycerol freezer stock and grown in 3 mL of media for 13–15 h. Cells were then diluted to OD600 = 3 × 10⁻⁴ with relevant inducers in a 500 µL volume in 24-well plates sealed with a tin foil adhesive (VWR, F96VWR100). Cells were grown to an OD600 = 0.3 (approximately 6 h) and placed on ice prior to measuring via flow cytometry with a FL1 gain of 600. All growth was conducted shaking at 250 rpm at 37 °C.

**Aspartate experiments.** Aspartate induction experiments were conducted in *E. coli* BW29655 (BW28357 Δ(*envZ-ompR*)520(::FRT); CGSC #7934; Yale University). M9 media containing 1× M9 salts (42 mM Na₂HPO₄, 24 mM KH₂PO₄, 8.9 mM NaCL, 19 mM NH₄Cl; Teknova, M1902), 2 mM MgSO₄ (VWR, BDH9246-500G), and 0.1 mM CaCl₂ (Alfa Aesar, L13191) were used with 22.2 mM glucose (Avantor, 4908-06) as a carbon source, and 2 g L⁻¹ casamino acids, 50 µg mL⁻¹ ampicillin, 35 µg mL⁻¹ chloramphenicol, 100 µg mL⁻¹ spectinomycin, 10 µM IPTG and 50 ng mL⁻¹ anhydrotetracycline (aTc; Takara Bio USA, 631310) were used. Then, 3 mL of this medium in a 14 mL culture tube was inoculated to OD600 = 5 × 10⁻³ from a single use 15% glycerol stock stored at −80 °C containing cells frozen during exponential phase. Bacteria were grown for 2 h shaking at 250 rpm at 37 °C. Amino acids were then removed by centrifuging at 3220 × g for 5 min, resuspending in 5 mL of media without casamino acids, centrifuging at 3220 × g for 5 min, and resuspending in 5 mL of media without casamino acids. Aspartate was added to the culture and bacteria were grown for 2 h shaking at 250 rpm at 37 °C, placed on ice, and then measured via flow cytometry with an FL1 gain of 750.

**Computational analysis of the phosphatase hot spot residue.** To estimate the fraction of known SKs that contain the phosphatase hot spot residue, we first assembled a library of non-redundant SK sequences from 4861 NCBI (National Center for Biotechnology Information) RefSeq bacterial genomes using HMMER3[62]. We used hmmsearch to identify all proteins that had a C-terminal kinase core composed of a single kinase domain (Pfam: HisKA, HisKA_2, HisKA_3, His_kinase, H-kinase_dim) followed by an HATPase_c domain (reporting threshold set to 12.0 for each). We eliminated SKs with non-canonical signaling architectures by requiring that each had at least a minimal sensing region (>10 a.a. N terminal of the kinase core) and contained neither a Receiver domain (Response_reg) nor a histidine phosphotransfer domain (Hpt). This constraint resulted in 105,144 SK proteins. To eliminate redundant sequences from this pool, we used usearch[63] to cluster the sequences according to a 60% sequence similarity threshold (using '-cluster_fast' and '-sort length' parameters). The centroids of each cluster were then used as representatives of non-redundant SKs, resulting in 56,855 proteins. We next created a hidden Markov model (HMM) representing the G2 box motif (Supplementary Fig. 9) by aligning 12 representative G2 box sequences[64] and using hmmbuild to create a model. This model was then used with hmmsearch (default parameters) to identify SKs in the non-redundant set that match, yielding 38,966 SKs with putative G2 box motifs. Two additional criteria were used to eliminate false positives: (1) the putative G2 box must align to the correct region of the protein (C terminal to the HisKA domain), and (2) the G2 box must have G3 and G5 present when aligned to the HMM. Applying these constraints left 36,508 SKs remaining, constituting 64.21% of the full non-redundant SK data set. Finally, the distribution of residues in the second position of the GXGXG motif were tabulated from these SKs.

**Tetrathionate and thiosulfate experiments.** Tetrathionate and thiosulfate induction experiments were conducted with *E. coli* BW28357 (CGSC#: 7991, Yale University). M9 media were used with 1 × M9 salts, with 43.4 mM glycerol (Fisher BioReagents, BP229-1) as a carbon source, 2 g L⁻¹ casamino acids (EMD Millipore, 2240-500GM), 35 µg mL⁻¹ chloramphenicol, and 100 µg mL⁻¹ spectinomycin. For thiosulfate experiments, 200 µM IPTG and 20 ng mL⁻¹ aTc were used, and leaky expression of the TtrSR TCS without inducers was found to be sufficient. Ligand induction was achieved with K₂S₄O₆ (Sigma-Aldrich, P2926-25G) or Na₂S₂O₃ (Sigma-Aldrich, 217247-25G). The experiment was started by inoculating 3 mL of media in a 14 mL culture tube to OD600 = 1 × 10⁻⁴ from a single-use 15% glycerol stock frozen during exponential phase and stored at −80 °C. Bacteria were grown at 37 °C shaking at 250 rpm for 4 h, placed on ice, and then measured via flow cytometry with a FL1 gain of 600.

**Soil nitrate experiments**. Soil experiments were conducted with *B. subtilis* 168 Δ*ydfHI*::*camR,mCherry* (iND77; Supplementary Fig. 12). CSE media with 0.3% xylose and 3 μM IPTG were used without antibiotics in all experiments. IPTG and xylose levels were selected to achieve a large fold change of both wild-type and C415R NarXL TCSs (Supplementary Fig. 12). Soil (Miracle Gro, All-Purpose Garden Soil) was prepared by removing large particles with a 1.75 mm strainer, transferring 0.1 g to a 14 mL culture tube, and adding $NaNO_3$ or fertilizer (Vigoro, All-Purpose Plant Food) which contains 1.03 M $NO_3^-$. The experiment was started with an overnight culture inoculated from a 15% glycerol freezer stock and grown in 3 mL of media for 18 to 22 h of shaking at 250 rpm. Cells were diluted to OD600 $= 3 \times 10^{-4}$ and grown to OD600 = 0.075–0.125 shaking at 250 rpm. Then, 250 μL of cells were added to the soil, vortexed for 5 s to mix, and centrifuged for 20 s to settle, resulting in 1 mL of damp soil. Cells were incubated in soil with no shaking for 2 h and then placed on ice. Next, 5 mL of cold phosphate-buffered saline was added and the samples were vortexed for 10 s to resuspend the cells. Particulates were allowed to settle for 2 min and then the supernatant was passed through Whatman #1 filter paper (Sigma, WHA10016508) to further remove particulates. Samples were then measured on the flow cytometer with a FL1 gain of 700 and FL3 gain of 850 thresholded at 45% FL3. All experiments were conducted at 37 °C.

**Flow cytometry and sfGFP fluorescence calculation**. Flow cytometry was conducted with a BD FACScan flow cytometer. The instrument employed blue (488 nm, 30 mW) and yellow (561 nm, 50 mW) solid-state lasers (Cytex) and a 510/21 nm filter (FL1) to measure GFP and a 650 nm long pass filter (FL3) to measure mCherry. For each sample, 10,000–20,000 events were collected at 500–2000 events per second within a forward scatter (FSC), side scatter (SSC) gate. Rainbow calibration beads from Spherotech, Inc. (cat. no. RCP-30-20A) were also collected each day at identical detector gain settings. Flow cytometry data were processed with FlowCal[65]. Events were selected by discarding the first 250 and last 100 time ordered events, a density gate was then applied to select the densest 10% of events (~1000–2000 events) in FSC/SSC space to specifically select bacterial cells (Supplementary Fig. 13). FL1 fluorescence was transformed into MEFL units using a standard curve created from the calibration beads measured on that day. The geometric mean of the population was used to calculate the fluorescence of each sample.

To calculate sfGFP fluorescence, measured bacterial autofluorescence (119 MEFL for *E. coli* and 150 MEFL for *B. subtilis*) was subtracted from total cellular fluorescence. Some samples were not significantly different from cellular autofluorescence, resulting in exaggerated fold change calculations $\left(\frac{\text{induced sample}-\text{autofluorescence}}{\text{uninduced sample}-\text{autofluorescence}}\right)$. Therefore, when calculating the fold change, if the sfGFP expression fell below the limit of detection (LOD $= 3 * \sigma_{\text{autofluorescence}}$; 16.6 MEFL for *E. coli* and 36.4 MEFL for *B. subtilis*) the LOD was used in place of the measured sfGFP value to calculate a lower bound of the fold change.

**Transfer function modeling and parameter estimation**. All transfer function data were fit to an activating Hill equation $\left(y = \text{low} + (\text{high} - \text{low})\frac{x^n}{K_{\frac{1}{2}}^n + x^n}\right)$ using the LmFit python package[66]. Here, *y* is the sfGFP fluorescence (MEFL), *x* is the concentration of inducer (μM), low is the sfGFP fluorescence at 0 μM inducer (MEFL), high is the maximum sfGFP fluorescence (MEFL), $K_{1/2}$ is the concentration of inducer that gives rise to half-maximal sensor activation (μM), and *n* is the Hill coefficient. All transfer functions were experimentally measured on three separate days. Replicate data points were combined into a single data set. This set was fit by the Hill equation. To fit both low and high sfGFP values well, the fit residuals at each data point were weighted by multiplying the residual by the inverse of the mean at that data point. The 95% confidence intervals of fit parameter values were calculated using the conf_interval function in LmFit, which executes the *F*-test. Fit parameters for all experiments in this study are shown in Supplementary Data 5.

**Code availability**. The code used to generate a model of a TCS is included as a supplementary file to this article.

**Data availability**. The datasets generated during and/or analyzed during the current study are available from the corresponding author on reasonable request. DNA sequences are available from GenBank and accession numbers can be found in Supplementary Data 3, 4.

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

## Acknowledgements

This work was supported by the ONR Young Investigator Award N00014-14-1-0487 and NSF CAREER 1553317. B.P.L. was supported by the DoD, Air Force Office of Scientific Research, National Defense Science and Engineering Graduate (NDSEG) Fellowship, 32 CFR 168a. We thank the Joel Moake lab for generous sharing of their flow cytometer.

## Author contributions

B.P.L. conceived of the project. J.J.T. supervised the project. B.P.L, N.D., and R.P. performed preliminary work and built DNA constructs. B.P.L. collected nitrate and aspartate data. R.P. collected tetrathionate and thiosulfate data. B.P.L. performed all data analyses. L.A.H. performed the bioinformatic analysis. B.P.L., L.A.H., and J.J.T. wrote the manuscript.

## Additional information

**Competing interests:** The authors declare no competing interests.

