## [Peer Review File · Nature Communications]

Reviewers' comments:

Reviewer #1 (Remarks to the Author):

Landry et al. designed a general strategy for improving sensitivity of the bacterial two-component systems (TCS) to the sensory stimuli via manipulating of the phosphatase activity of the sensory kinase/phosphatase (SK) protein. They also identified a conserved amino acid residue amenable for manipulating phosphatase activity in proteins that have not been experimentally characterized. To demonstrate the utility of their general approach, the authors have engineered an improved nitrate sensing TCS in the soil bacterium *Bacillus subtilis* and showed its ability to sense a broad range of nitrate levels in the soil samples containing a commercial fertilizer. This study represents a significant advance in synthetic biology expected to broaden engineering of TCSs to diverse practical applications. The ideas described here are clever and innovative, the experiments have been performed systematically and meticulously, and the writing style is logical and clear. I have no major concerns.

There are a few minor issues:

- Line 9: TCSs are one of the largest families of regulatory systems in bacteria but, to my knowledge, not the largest. Single component regulators are more abundant.
- Line 46: There are also SKs that sense gases (O₂, NO, etc)
- Suppl. Note 1:

The numbers of equations are shifted; submit the Suppl. as a pdf file.

Use subscript for 'p' and 't' throughout; replace 'u' with 'micro'.

Simplify the last sentence: "However, when the phosphatase..."

Reviewer #2 (Remarks to the Author):

This is an interesting manuscript that describes the engineering of two-component systems as biosensors with altered detection thresholds. The modular architecture of two-component systems enables the engineering of different biosensors but the dose response is often limited by the intrinsic activities of individual system. The authors developed a simple and clever strategy to alter the detection threshold by using mutants with impaired phosphatase activity. There is an abundance of experimental data demonstrating the application of this strategy to different systems. The manuscript presents strong evidence that mutations in the conserved histidine kinase domain can tune the dose response for several different systems. However, mutations, especially the one in the G2 box suggested as the hot spot residues, may have complex effects on not only the phosphatase but also the kinase activity, thus cautions should be exercised to attribute the effect to the phosphatase activity. Nevertheless, the strategy could be valuable for synthetic biology efforts of designing novel biosensors. The following are a few minor issues that need to be addressed:

1. It is obvious that the RR phosphorylation level is determined by the ratio of the kinase and phosphatase activity. An increase of kinase activity is equally possible as a decrease of phosphatase activity to modulate RR phosphorylation thus alter the input-output function and the detection threshold. The entire manuscript appears to focus solely on the phosphatase and does not recognize potential mutational effects on the kinase activity.

This is especially true for mutants with substitutions in the G2 box (Taz T436, TtrS L627 and ThsS L547). The G2 box flanks the ATP lid that is important for binding to ATP for the kinase activity or ADP for the phosphatase activity. Without well-characterized autokinase activities of these mutants, gene reporter assay cannot distinguish whether the mutation affects the kinase or the phosphatase activity. Should clarify to include this possibility.

The kinase and the phosphatase activities have reciprocal effects on system output. The authors model the input as the ranging value of the autokinase rate k_k . This naturally leaves the phosphatase activity as the factor to adjust the input-output function. Vice versa, if the input is modeled as the changing of the phosphatase rate k_p , then the kinase rate can affect the output dose response as well as the detection threshold.

2. Line 74: Is "comprising" a typo for "compromising"?

3. Line 100-101: "C415 targets the DHP heterodimerization interface" The word heterodimerization is a little misleading and not commonly used to refer the interaction interface between DHP and RR.

4. Line 104-105 and Fig. 2: I assume that YdfI is fused with NarL to function as a transcription reporter in *B. subtilis*. The reason of why YdfI is selected is not well explained. Does nitrate or nitrite affect the transcription of the YdfI-regulated promoter ydfJ?

5. Line 111: "SK and RR concentrations were re-optimized" It is not clear in the text why they were re-optimized or re-optimized for what. The corresponding section in Supplementary Line 107 is also not clear. What does the "insensitive region of NarL-YdfI induction" refer to? More explanation may help.

6. Line 128: "moderate detection threshold changes have little effect on dynamic range" This conclusion is not consistent with data in Supplementary Fig 5. Except for the last data point, it appears that any decrease of detection threshold will cause corresponding reduction of the dynamic range.

7. Supplementary Fig. 5b: Is the color coding reversed? Data points colored in darker blue indicate higher fractions of WT NarX and lower detection thresholds, not consistent with data in Fig. 3.

8. Line 162: The GXGXG motif is also involved in binding to ATP, which is important for the kinase activity.

We thank the reviewers for carefully reading our manuscript and providing useful comments. We have addressed each comment, and included a point-by-point response below.

Reviewer #1

Landry et al. designed a general strategy for improving sensitivity of the bacterial two-component systems (TCS) to the sensory stimuli via manipulating of the phosphatase activity of the sensory kinase/phosphatase (SK) protein. They also identified a conserved amino acid residue amenable for manipulating phosphatase activity in proteins that have not been experimentally characterized. To demonstrate the utility of their general approach, the authors have engineered an improved nitrate sensing TCS in the soil bacterium Bacillus subtilis and showed its ability to sense a broad range of nitrate levels in the soil samples containing a commercial fertilizer. This study represents a significant advance in synthetic biology expected to broaden engineering of TCSs to diverse practical applications. The ideas described here are clever and innovative, the experiments have been performed systematically and meticulously, and the writing style is logical and clear. I have no major concerns.

Line 9: TCSs are one of the largest families of regulatory systems in bacteria but, to my knowledge, not the largest. Single component regulators are more abundant.

The reviewer is correct, and we agree that our original language may have caused confusion. We have revised the sentence to indicate that TCSs are the largest family of multi-step signal transduction pathways in biology.

Line 46: There are also SKs that sense gases (O₂, NO, etc)

Yes, gases are an important class of molecules sensed by TCSs and we thank the reviewer for pointing this out. We have now added a statement that TCSs also sense gases and citation.

Suppl. Note 1: The numbers of equations are shifted

Thank you for noticing this error. We have corrected it.

submit the Suppl. as a pdf file.

We have uploaded a pdf file of the supplementary information

Use subscript for 'p' and 't' throughout; replace 'u' with 'micro'.

For the latter comment, we have replaced all instances of “uM” with “μM” in Supplementary Note 1. Thank you for noticing this mistake.

Additionally, we have replaced an incorrectly written “k_p” with “k_k” in equation 11. However, in the former comment, we are unsure if the reviewer is referencing this typo, or if there was general confusion over the use of k_k, k_t, k_{-k} and k_p. Here, k_k refers to

autokinase activity of the SK, and k_t refers to kinase activity of the SK~P on the RR. These are two different reactions with different constants. Likewise, k_{-k} refers to autophosphatase activity of the SK and k_p refers to phosphatase activity of the SK on the RR~P. This notation follows the original notation by Batchelor and Goulian which we have preserved in this publication.

Simplify the last sentence: "However, when the phosphatase..."

Thank you, we have simplified it in the revised manuscript.

Reviewer #2

This is an interesting manuscript that describes the engineering of two-component systems as biosensors with altered detection thresholds. The modular architecture of two-component systems enables the engineering of different biosensors but the dose response is often limited by the intrinsic activities of individual system. The authors developed a simple and clever strategy to alter the detection threshold by using mutants with impaired phosphatase activity. There is an abundance of experimental data demonstrating the application of this strategy to different systems. The manuscript presents strong evidence that mutations in the conserved histidine kinase domain can tune the dose response for several different systems. However, mutations, especially the one in the G2 box suggested as the hot spot residues, may have complex effects on not only the phosphatase but also the kinase activity, thus cautions should be exercised to attribute the effect to the phosphatase activity. Nevertheless, the strategy could be valuable for synthetic biology efforts of designing novel biosensors. The following are a few minor issues that need to be addressed:

It is obvious that the RR phosphorylation level is determined by the ratio of the kinase and phosphatase activity. An increase of kinase activity is equally possible as a decrease of phosphatase activity to modulate RR phosphorylation thus alter the input-output function and the detection threshold. The entire manuscript appears to focus solely on the phosphatase and does not recognize potential mutational effects on the kinase activity.

This is especially true for mutants with substitutions in the G2 box (Taz T436, TtrS L627 and ThsS L547). The G2 box flanks the ATP lid that is important for binding to ATP for the kinase activity or ADP for the phosphatase activity. Without well-characterized autokinase activities of these mutants, gene reporter assay cannot distinguish whether the mutation affects the kinase or the phosphatase activity. Should clarify to include this possibility.

The kinase and the phosphatase activities have reciprocal effects on system output. The authors model the input as the ranging value of the autokinase rate k_k . This naturally leaves the phosphatase activity as the factor to adjust the input-output function. Vice versa, if the input is modeled as the changing of the phosphatase rate k_p , then the kinase rate can affect the output dose response as well as the detection threshold.

The reviewer makes several important points here. First, we agree that modulating SK kinase activity should also enable tuning of TCS detection threshold, in a manner reciprocal to that of modulating phosphatase activity. To address this point, we have performed additional modeling of the effect of changes in kinase activity on detection

threshold. Our results, shown in Supplementary Note 1 and Supplementary Figure 1, confirm that increasing kinase activity decreases detection threshold (and vice-versa). However, our primary goal in this study is to increase TCS sensitivity (i.e. decrease detection threshold) rather than to decrease it. To achieve this goal, one can use mutations that decrease phosphatase activity or increase kinase activity. Based on previous mutational studies in the literature, we believe it is much easier to identify mutations that do the former than the latter. Thus, for practical purposes, we chose to focus our study on decreasing SK phosphatase activity. We have added a new paragraph discussing these new modeling results and why we chose to focus on phosphatase activity in the main text.

Second, we agree that with gene expression assays alone, we cannot distinguish whether SK mutations decrease phosphatase activity or increase kinase activity. We have edited the main text to state both possibilities for mutants for which *in vitro* enzymatic activities have not been measured (NarX(D558V), and all of the TtrS and ThsS mutants). However, we note that for the Taz T436 mutants mentioned by the reviewer, their phosphatase activity has been shown to be decreased *in vitro* and, although the data is less clear, their kinase activity does not appear to have been increased (we are referring to EnvZ T402 in Zhu and Inuoye 2002; this is the equivalent residue to Taz T436 since the two proteins share a cytosolic domain).

2. Line 74: Is “comprising” a typo for “compromising”?

Thank you for pointing out this typo, we have fixed it.

3. Line 100-101: “C415 targets the DHp heterodimerization interface” The word heterodimerization is a little misleading and not commonly used to refer the interaction interface between DHp and RR.

Thank you. We have changed “heterodimerization interface” to “interaction interface” throughout the text.

4. Line 104-105 and Fig. 2: I assume that YdfI is fused with NarL to function as a transcription reporter in *B. subtilis*. The reason of why YdfI is selected is not well explained. Does nitrate or nitrite affect the transcription of the YdfI-regulated promoter ydfJ?

Yes, we fused the REC domain of NarL to the DBD of YdfI to connect the *E. coli* NarXL TCS to a robust transcriptional output (P_{ydfJ115} promoter) in *B. subtilis*. We selected the YdfI DBD because it shares high homology with NarL, and because YdfI regulates a single promoter, P_{ydfJ}, in *B. subtilis*, which should prevent our nitrate sensor from inducing off-target transcription. Additionally, P_{ydfJ} is well characterized and nitrate and nitrite do not affect its activity. We have added a new sentence to the legend of Supplementary Figure 2 to discuss this design decision.

5. Line 111: “SK and RR concentrations were re-optimized” It is not clear in the text why they were re-optimized or re-optimized for what. The corresponding section in Supplementary Line

107 is also not clear. What does the “insensitive region of NarL-YdfI induction” refer to? More explanation may help.

When measuring detection thresholds in Figure 2, we selected a single set of SK and RR expression levels that resulted in a large dynamic range of all three sensors. We have revised the caption of Supplementary Fig. 3 to succinctly describe this fact.

Furthermore, we found that the optimal fold change of the wild-type and mutant TCSs occurred at different expression levels (Supplementary Fig. 3). Therefore, when comparing the dynamic range, we compared the maximal dynamic range of each TCS when their expression levels were individually optimized. We have revised lines 109-112 in the text to make this clearer.

6. Line 128: “moderate detection threshold changes have little effect on dynamic range” This conclusion is not consistent with data in Supplementary Fig 5. Except for the last data point, it appears that any decrease of detection threshold will cause corresponding reduction of the dynamic range.

We agree that most any decrease in detection threshold does cause a reduction in dynamic range. However, we were attempting to convey that this trade-off is not severe unless the detection threshold change is very large. For example, in Supplementary Fig. 5, the data show an eight-fold decrease in detection threshold with only a two-fold decrease in the dynamic range. This result is recapitulated in our model predictions (Supplementary Note 1; Supplementary Figs. 1, 5). On the other hand, for the 100-fold decrease in detection threshold achieved between 100% and 0% NarX:NarX(C415R), dynamic range decrease from 2334-fold to 8-fold. We have added a discussion of these trade-offs in the main text.

7. Supplementary Fig. 5b: Is the color coding reversed? Data points colored in darker blue indicate higher fractions of WT NarX and lower detection thresholds, not consistent with data in Fig. 3.

Yes, it was reversed. We thank the reviewer for noticing this error, and have corrected it.

8. Line 162: The GXGXG motif is also involved in binding to ATP, which is important for the kinase activity.

Thank you, this is an important point to mention. We have added in details mentioning how the GXGXG motif binds ATP for the kinase reaction.

REVIEWERS' COMMENTS:

Reviewer #2 (Remarks to the Author):

The revised manuscript has addressed all the questions raised previously and I recommend it for publication. The following are two minor issues.

Line 17: "our method to the majority of TCSs." I feel this may be an overstatement. This work certainly demonstrates that the TCS tuning method can be applied to several systems. However, given the great diversity of TCSs, whether the majority of TCSs share the same phosphatase-tuning mechanism remains to be explored.

Line 122-123: I understand that the ratio of maximal to minimal output is commonly used to calculate the dynamic range. But this calculation might be a little misleading here. The fold change of output for the WT is 1909, much greater than the 78-fold for C415R. But the absolute difference between the maximal and basal output shown in Figure 2b indicates a much larger amplitude range for C415R. In this case, C415R might be a better sensor than WT in both sensitivity and amplitude range. The fold change is greatly exaggerated for the WT due to the low basal level. I'm wondering whether the fold change is the best indicator of the dynamic range. A little clarification may help.

Reviewer #2

Line 17: "our method to the majority of TCSs." I feel this may be an overstatement. This work certainly demonstrates that the TCS tuning method can be applied to several systems. However, given the great diversity of TCSs, whether the majority of TCSs share the same phosphatase-tuning mechanism remains to be explored.

We agree and have eliminated this statement from the relevant sentence.

Line 122-123: I understand that the ratio of maximal to minimal output is commonly used to calculate the dynamic range. But this calculation might be a little misleading here. The fold change of output for the WT is 1909, much greater than the 78-fold for C415R. But the absolute difference between the maximal and basal output shown in Figure 2b indicates a much larger amplitude range for C415R. In this case, C415R might be a better sensor than WT in both sensitivity and amplitude range. The fold change is greatly exaggerated for the WT due to the low basal level. I'm wondering whether the fold change is the best indicator of the dynamic range. A little clarification may help.

Thank you for this great insight! We completely agree that in many circumstances, the amplitude range of a sensor may be more important than the dynamic range. Analysis of our data shows that the amplitude ranges of NarXL (C415R) and (D558V) are indeed ~3 times larger than that of wild-type NarXL. Furthermore, we also see an increase in amplitude range of the thiosulfate and tetrathionate sensors when we decrease their detection thresholds as well. We have added new text discussing this fact and the potential value of increased amplitude range.